# His108Arg Transthyretin Amyloidosis—Shedding Light on a Distinctively Malignant Variant

**DOI:** 10.3390/jcm13247857

**Published:** 2024-12-23

**Authors:** Christina Binder, Lena Marie Schmid, Christina Kronberger, Michael Poledniczek, René Rettl, Johanna Schlein, Nikita Ermolaev, Luciana Camuz Ligios, Michaela Auer-Grumbach, Christian Hengstenberg, Roza Badr Eslam, Johannes Kastner, Jutta Bergler-Klein, Andreas Anselm Kammerlander, Franz Duca

**Affiliations:** 1Department of Internal Medicine, Division of Cardiology, Medical University of Vienna, 1090 Wien, Austria; christina.binder@meduniwien.ac.at (C.B.); lena.schmid@meduniwien.ac.at (L.M.S.); christina.kronberger@meduniwien.ac.at (C.K.); rene.rettl@meduniwien.ac.at (R.R.); nikita.ermolaev@meduniwien.ac.at (N.E.); luciana.camuzligios@meduniwien.ac.at (L.C.L.); christian.hengstenberg@medunwien.ac.at (C.H.); roza.badr-eslam@meduniwien.ac.at (R.B.E.); johannes.kastner@meduniwien.ac.at (J.K.); jutta.bergler-klein@meduniwien.ac.at (J.B.-K.); andreas.kammerlander@meduniwien.ac.at (A.A.K.); 2Department of Cardiac Surgery, Medical University of Vienna, 1090 Wien, Austria; johanna.schlein@meduniwien.ac.at; 3Department of Orthopedics and Trauma Surgery, Medical University of Vienna, 1090 Wien, Austria; michaela.auer-grumbach@medunwien.ac.at

**Keywords:** amyloidosis, cardiac amyloidosis, transthyretin, His108Arg, ATTRv, heart failure, arrhythmia, outcome, transplantation

## Abstract

Variant transthyretin amyloidosis cardiomyopathy (ATTRv-CM) is a rare form of cardiac amyloidosis associated with many possible mutations in the transthyretin gene, presenting as various distinct clinical phenotypes. Among these, the His108Arg mutation is the most prevalent TTR variant in Austria. However, data describing its clinical phenotype are lacking. This study aims to describe the characteristics, clinical manifestations, and outcomes of patients with the His108Arg variant focusing on cardiac involvement, disease progression, response to therapy, and imaging findings. **Methods:** Patients were enrolled from a prospective cardiac amyloidosis registry. The baseline assessment included comprehensive echocardiography, cardiac magnetic resonance imaging, a biomarker analysis, and a clinical evaluation. Patients were followed longitudinally, with outcomes such as arrhythmias, heart failure hospitalizations, and response to disease-targeted therapies recorded. **Results:** Between March 2012 and June 2024, a total of 20 carriers of the His108Arg variant were identified, with 12 exhibiting clear cardiac involvement and 8 remaining asymptomatic. The median age at diagnosis was 62.3 years with significant heterogeneity in the clinical presentation. Patients with ATTRv-CM had a high prevalence of atrial and ventricular arrhythmias, a reduced left ventricular ejection fraction, and elevated cardiac biomarkers. The majority received specific disease-modifying therapies, with varying tolerance and responses. A longitudinal follow-up indicated frequent arrhythmic events, heart failure exacerbations, and three cases of heart transplantation, underscoring the need for stringent monitoring and individualized management strategies. **Conclusions:** This study represents a unique, comprehensive analysis of the His108Arg variant in ATTR-CM, highlighting its clinical heterogeneity and significant impact on cardiac function and clinical outcomes.

## 1. Introduction

Transthyretin amyloid cardiomyopathy (ATTR-CM) has been increasingly recognized as an important cause of heart failure (HF) associated with a hypertrophic-restrictive phenotype. Transthyretin (TTR) is almost exclusively produced in the liver and in its physiologic form is a transport protein for retinol and thyroxin (T4). Misfolded TTR is unstable in its configuration and can dissociate from its regular tetrameric form to monomers and form amyloid fibrils which accumulate in tissues with varying distribution in different organs (Figure 1). While the exact etiology of wild-type ATTR-CM is not clear, it is associated with advanced age and is found predominantly in elderly males. In contrast, variant forms of ATTR-CM (ATTRv) vary with respect to the age of onset, organ involvement, and the clinical course, depending on the distinct disease-causing genetic point mutation.

The most prevalent TTR variants that cause a cardiomyopathy phenotype include V30M (Val30Met), which is endemic in Portugal, Sweden, and Japan, and V122I (Val122Ile), frequently observed among African–American populations. V30M mutations primarily present as a neurological phenotype with early-onset cases often exhibiting severe peripheral and autonomic polyneuropathy, while late-onset cases may present with a mixed phenotype, including cardiac involvement. In contrast, the V122I mutation is primarily linked to a cardiac phenotype with a high risk of HF and arrhythmias. Other mutations, such as T60A (Thr60Ala) and Y114C (Tyr114Cys), also demonstrate variable organ involvement, contributing to significant morbidity and mortality [1,2]. The mutation type influences disease progression, with earlier-onset mutations generally associated with a more aggressive course and poorer outcomes, underscoring the need for mutation-specific management and therapeutic approaches in ATTRv.

In Austria, a cluster of nine patients with His108Arg mutations has previously been identified in Vienna (Eastern Austria) [3]. Overall, very few studies or case series have described this mutation. A single case was described in a population of ATTR patients with neurological and cardiac involvement in Singapore [4] and seven individuals (six patients with cardiomyopathy and one carrier) were found in a Swedish cohort [5]. Despite the small number of patients, it seems that the His108Arg mutation can present with varying degrees of polyneuropathy and cardiomyopathy. However, the scarcity of data and experience makes it difficult to predict the clinical course and treatment response to TTR-specific disease-modifying therapies in patient with His108Arg ATTR-CM, as no data currently exist on this specific variant [5].

We aim to characterize a well-defined population of patients from an Austrian amyloidosis registry with a mutation in the His108Arg gene locus causing the variant ATTR-CM by describing the clinical course including multimodality cardiac imaging to assess structural changes, as well as responses to ATTR-specific treatments.

## 2. Materials and Methods

### 2.1. Study Population and Design

Patients with ATTR-CM were included in this study from our dedicated cardiac amyloidosis (CA) outpatient clinic at the department of Cardiology at the General Hospital of Vienna, a university-affiliated, tertiary care, and reference center for CA in Vienna Austria. Patients presenting to our outpatient CA clinic were either referred from in-house patient wards or outpatient clinics of other departments, or from other centers for further diagnostic testing or initiation of CA-specific therapies. All patients gave written informed consent and the clinical registry has previously been approved by the local ethics committee of the Medical University of Vienna (EK No (#1079/2023) and was conducted in accordance with the ethical guidelines stated in the Declaration of Helsinki.

### 2.2. Diagnosis of Cardiac Amyloidosis

The diagnosis of ATTR was made according to current expert consensus and guidelines [6,7,8]. All patients underwent ^99^mTc-3,3-diphosphono-1,2-propanodicarboxylic acid (DPD) bone scintigraphy and laboratory testing in order to exclude light-chain amyloidosis. The diagnosis of ATTR -CM was made when scintigraphy showed a positive tracer uptake (Perugini Grade II or III) and presence of a paraprotein was excluded in serum or urine samples. Cardiac or other organ biopsies were only performed when non-invasive diagnostic testing remained non-diagnostic.

### 2.3. Genetic Testing and Family Screening

In accordance with Austrian legislation, genetic counseling and testing were offered to all patients diagnosed with ATTR amyloidosis. Genomic DNA was extracted from patients’ whole blood samples using the QIAsymphony^®^ system (Qiagen, Hilden, Germany). Exons 1–4 of the transthyretin gene were amplified using Phusion Green Hot Start II PCR Master Mix (Thermo Fisher) with an annealing temperature of 60 °C. After digesting residual primers with Exo SAP-IT Express (Thermo Fisher Scientific, Atlanta, GA, USA), sequencing was carried out using BigDye Terminator Mix v3.1 (Thermo Fisher) on an Applied Biosystems 3130xl Genetic Analyzer. Transthyretin gene sequences (NM_000371.3; NG_009490.1) were analyzed with SeqScape v4.0 and DNADynamo software version 1.0.

When a transthyretin-specific mutation was identified in an index patient, genetic testing and clinical cardiac screening including cardiac imaging and biomarker assessment was offered to all family members over the age of 18.

### 2.4. Clinical Work-Up and Standard of Care

Baseline assessment included a detailed clinical and family history, a physical exam, 6 min walk test, laboratory sampling, and cardiac biomarkers, as well as a comprehensive transthoracic echocardiography study including 2D speckle tracking to quantify left-ventricular global longitudinal strain (LV-GLS) when imaging quality allowed (GE Healthcare, Chicago, IL, USA). Contrast-enhanced cardiac magnetic resonance imaging (CMR) including T1-mapping was performed in patients without contraindications for evaluation of cardiac dimensions, functions, and assessment of fibrosis and amyloid burden as previously described (Siemens, Munich, Germany) [9,10].

Patients were scheduled for follow-up visits in 6- to 12-month intervals or as deemed clinically appropriate by the attending CA specialist. These visits included a comprehensive clinical evaluation and follow-up echocardiography. Increased visit frequency was implemented when signs or symptoms of clinical deterioration were evident.

### 2.5. Definition of Outcomes

The primary clinical outcomes were defined as cardiac death and worsening of HF. Cardiac death was classified as any death directly attributable to cardiac causes, including arrhythmic events, progressive HF, or sudden cardiac arrest, in line with established criteria. Worsening of HF was characterized by the need for hospitalization due to HF symptoms, an escalation in diuretic treatment, or a documented decline in functional status requiring medical intervention. In patients who had an intracardiac implantable defibrillator device (ICD) at baseline or received such a device during follow-up, we also assessed the number and appropriateness of shocks from the respective device during the observation period. Appropriate ICD-shocks were defined as a therapeutic intervention delivered by the device in response to detected life-threatening ventricular arrhythmias, such as ventricular tachycardia or ventricular fibrillation. 

These endpoints were chosen to capture clinically meaningful adverse events that reflect the progression of disease and its impact on patient survival and quality of life.

### 2.6. Statistical Analysis

Acknowledging the small patient population, statistical analysis in this study was primarily descriptive while aiming to provide an overview of baseline characteristics, clinical outcomes, and imaging findings among patients with the His108Arg mutation in the transthyretin gene. Categorical variables were summarized as frequencies and percentages, while continuous variables were presented as medians with interquartile ranges (IQR) to account for non-normal distributions. To assess changes in parameters over time, paired *t*-tests or Wilcoxon signed-rank tests were used to analyze the change in clinical and imaging parameters between the first visit and the last visit at our center. In patients who received a cardiac transplantation (HTX), the last visit before HTX was used for analysis of clinical and structural parameters. The level of statistical significance was set to <0.05 and SPSS (version 29.0) was used for all calculations and analyses.

## 3. Results

Between March 2012 and June 2024, a total number of 387 individuals were diagnosed with ATTR amyloidosis at our center. A mutation in the TTR-gene was detected in 47 (12.1%) and of these, a mutation in the His108Arg gene locus was identified in 20 individuals (5%). However, while 12 patients had a positive cardiac phenotype, the remaining 8 individuals showed no clear sign of cardiac or other clinical manifestation of amyloid-associated disease (Figure 2).

### 3.1. Patient Population and Mode of Diagnosis

In patients with His108Arg-associated ATTR-CM, the median age at diagnosis was 62.3 years (IQR 58.6–65.5). The ATTR-CM population consisted of eight males and four females, while the eight individuals with a verified His108Arg mutation without definite cardiac manifestations were five females and three males. Notably, we identified two family clusters; Patients 2, 11, and 12 were related, as well as Patients 6, 7, and 8 (Appendix A). The patient characteristics at the time of the baseline evaluation are shown in Table 1.

A myocardial biopsy was performed in three patients and a non-cardiac biopsy in conjunction with the evidence of cardiac involvement in cardiac imaging was used to confirm ATTR in one patient. In all others, non-invasive diagnostics using a DPD bone scintigraphy, transthoracic echocardiography, and/or CMR were sufficient to establish the diagnosis. Paraproteinemia was excluded in all patients. Of note, the timepoint of ATTR diagnosis did not always correspond to the time of the baseline evaluation at our institution, as some patients were referred to our amyloidosis reference center at a later time. The median time from diagnosis to the baseline evaluation was 0.0 months (IQR 0.0–2.4), with a large range of up to 68.0 months in one patient with a primarily neurological presentation, in whom the first suspicion of cardiac involvement was raised in 2014 by an echocardiography, but who was not referred to our CA clinic for a baseline cardiac evaluation until 2019.

### 3.2. Patient Characteristics at the Time of Clinical Diagnosis

At the time of the baseline evaluation, the majority of patients with ATTR-CM had symptoms of HF and markedly elevated cardiac biomarkers (median troponin T 53 ng/L (IQR 28–82) and median NT-proBNP 2033 pg/mL (IQR 1278–3572)). The median 6 min walk distance was 407 m (IQR 357–425) at initial presentation to our center. One patient had significant coronary artery stenosis and required elective stenting of the left anterior descending artery during follow-up. A diagnostic coronary angiography showed coronary aneurysms in a further patient, without evidence of significant stenosis. At the baseline, atrial fibrillation or an atrial flutter had been diagnosed in four patients and three had previously received a cardiac implantable electronic device (CIED). Of these, one patient had an ICD after a survived episode of sudden cardiac death (SCD), and the other had received a cardiac resynchronization therapy defibrillator device (CRT-D) due to symptomatic sick sinus syndrome with concurrent HF with a reduced ejection fraction and left bundle-branch block. Four patients were on betablockers, seven on angiotensin-converting enzyme inhibitors or angiotensin blockers, eight were regularly taking mineralocorticoid antagonists, and two patients were on daily sodium–glucose transporter-2 inhibitor therapy. At the time of the baseline evaluation, the median daily cumulative dose of the loop diuretics was 30 mg of furosemide or a furosemide equivalent per day. Polyneuropathy (PNP) was confirmed in five patients at the time of the baseline evaluation. In only one patient, PNP had been diagnosed prior to the diagnosis of ATTR-CM (Patient 5). This patient was significantly impaired by PNP symptoms and the neurological diagnosis prompted a cardiac work-up. The remaining patients with evidence of PNP on nerve conduction testing experienced only mild symptoms. A description of the neurological manifestations is shown in Appendix A.

Table 2 shows an overview of all of the parameters at the baseline and throughout the follow-up.

### 3.3. Manifestations of Heart Failure and Structural Cardiac Changes at Baseline and Follow-Up

At the time of the baseline evaluation, patients presented with pronounced left ventricular hypertrophy (interventricular septum thickness 21 mm (IQR 18–26) and a left ventricular mass of 410 g (IQR 356–490). The median left ventricular ejection fraction (LVEF) measured by an echocardiography was mildly reduced at 48% (37–61) and LV-GLS was already markedly impaired [−11.7%, IQR −13.5–(−10.0)] (Figure 3).

The Doppler findings were consistent with the elevated filling pressures, as demonstrated by a high E/e’ ratio (18.4, IQR 9.6–21.5), and patients had enlarged left atria. LVEF measured by CMR was slightly higher than that measured by the echocardiography (53% (IQR 47–59), which might be explained by the fact that patients with lower LVEF had previously received an ICD and were, therefore, not included in the CMR-cohort. As typical for CA, pre-contrast T1 times were significantly prolonged and the extracellular volume fraction (ECV) was elevated (68.4%, IQR 43.5–69.9).

Compared to the baseline, we found a significant rise in the median NT-proBNP levels from 2033 to 3183 pg/mL (Z-score 2.76, *p* = 0.006) at the time of the last recorded visit, while there was only a slight increase in troponin levels from 53 to 69 ng/L (Z-score 1.21, *p* = 0.225). During the course of the follow-up, the interventricular septum thickness measured by the echocardiography increased, but the posterior wall thickness remained constant. Of note, there was a progressive decline in LV-GLS from −11.7% [IQR −13.5–(−10.0), *p* = 0.003], while overall, no significant change in LVEF was noted (Z-score −0.45, *p* = 0.656). However, these data may be influenced by the follow-up data of one patient (Patient 4) in whom LVEF increased from 25% to 56% after the implantation of a CRT-D device (Figure 4 panel C). The parameters describing left ventricular filling pressures and right ventricular size and function remained unchanged. However, our data indicate a significant increase in tricuspid regurgitation severity despite stable systolic pulmonary artery pressures.

### 3.4. Follow-Up of Arrhythmias

The development of atrial and ventricular arrhythmias was frequent among the observed population. A CIED was implanted in all but two patients during the follow-up period of 4.6 years (IQR 3.0–8.0). The indication for ICD implantation was made according to current guidelines for SCD prevention, in conjunction with an individualized risk assessment based on family history and amyloid burden by CMR, while also considering patient preference. One patient survived an episode of SCD and received an ICD for secondary prevention, whereas the indication for ICD implantation was primary prevention in all other patients. Table 3 shows an overview of the CIED types which were implanted, as well as the indications and total follow-up time. Of the nine patients with an ICD, only two patients received at least one shock during follow-up, of which only one was inappropriate.

Nine out of the twelve patients with ATTRv-CM developed atrial fibrillation (AF, median time from ATTR diagnosis to AF onset 17.2 (IQR 13.0–32.9) months), which was associated with at least moderate symptoms in all patients. Electrical cardioversion was performed in four patients, but AF recurred in all patients during follow-up (Table 4). AF was especially problematic in one female patient (Patient 1) in whom AF episodes were recurrent despite three electric cardioversions and two ablation therapies and even led to an inappropriate ICD shock caused by a supraventricular tachycardia with a rate in the V2-zone (Patient 1). Due to significant left atrial enlargement and extensive atrial myopathy on electrophysiological mapping, atrioventricular–nodal ablation was finally performed as a last-resort option. Due to further clinical deterioration, the patient was listed for HTX.

Bradyarrhythmias were documented in four patients and were the indication for the implantation of a CIED in three of these patients. Interestingly, one of these patients received a pacemaker due to a high-degree atrioventricular block, which occurred 2 years after HTX. Notably, the high rate of ICD implantation (both primary and secondary) makes it impossible to accurately determine the true incidence of bradyarrhythmias within the study population.

### 3.5. Medical Management and TTR-Specific Treatment Regimens

The differences in therapy regimes were largely dictated by the approval and authorization of respective medications by regulations and reimbursement strategies of the local insurance companies, which were re-evaluated continuously on a case-by-case basis in the light of the increasing amount of data in the field of ATTR-CM and evidence for the clinical benefit of novel therapies. All patients except one, in whom Inotersen was initiated as a first-line therapy due to significant polyneuropathy and patient preference, received Tafamidis during their treatment at our amyloidosis center.

Of the six patients who were prescribed Inotersen, four patients discontinued the injections due to adverse effects. Interestingly, all patients initially tolerated the injections well, but developed side-effects including chills or fevers as well as gastrointestinal side-effects over the course of several injections which led to a discontinuation of Inotersen. Patient 9 even experienced a severe reaction which required hospitalization immediately following an injection after 7 months of therapy. Patient 6 developed thrombopenia and recurrent nose bleeds and bruising, which resolved after Inotersen was suspended. Of note, this patient had significantly impaired renal function and a creatinine clearance of 22 mL/min/m^2^. The only patient who tolerated Inotersen well was eventually switched to Patisiran due to disease progression (Patient 2).

### 3.6. Patient Outcomes

After a median follow-up time of 4.6 years (IQR 3–8.0), four patients died and three underwent HTX. Of these, two deaths were attributable to end-stage cardiac failure, while the third was non-cardiac and a result of fatal sequalae from a fall (Patient 3). Instances of the worsening of HF occurred in nine patients, with frequencies ranging from one to six episodes (Table 4). Of note, Patient 7, who suffered from recurrent cardiac decompensations, developed significant tricuspid regurgitation over the course of the follow-up period and is currently being evaluated for a transcatheter edge-to-edge repair of the tricuspid valve.

HTX was performed 6.7, 6.0, and 3.6 years after the initial ATTR-CM diagnosis in Patients 1, 2, and 10, respectively. The post-HTX hospitalization duration was 38 and 11 days in the two patients who survived after surgery. In Patient 10, a high-degree atrioventricular block occurred during HTX, necessitating dual pacing. The clinical picture raised concerns of severe primary graft dysfunction, prompting mechanical circulatory support and a surgical revision of suspected pulmonary artery stenosis. Despite these interventions, the patient experienced progressive multiorgan failure, coagulopathy, and significant bleeding complications, ultimately succumbing to these complications. Figure 5 shows an overview of the clinical timeline of each patient included in this analysis.

### 3.7. Phenotype-Negative His108Arg Mutation Carriers

In addition to the 12 patients with evident ATTR-CM, we also identified 8 individuals with a His108Arg mutation in the TTR gene who were asymptomatic and did not show any elevation in cardiac biomarkers or evidence of ATTR-CM on the transthoracic echocardiography or CMR. While the majority of these phenotype-negative subjects also had no tracer uptake on the DPD bone scintigraphy, we found that two of them indeed showed the tracer uptake of either Perugini grade 1 or 2 (Appendix A). Figure 6 shows the imaging results of a 59-year-old His108Arg carrier without clear evidence of ATTR-CM but a mild tracer uptake (Perugini grade II) on the DPD bone scintigraphy.

## 4. Discussion

In the present study, we describe the characteristics and clinical course of a population with the His108Arg variant in the transthyretin gene, causing ATTRv-CM. We found that this mutation was frequently associated with atrial and ventricular arrythmias, as well as HF hospitalizations, resulting in significant morbidity and a reduction in the quality of life, resulting in a high rate of CIED implantations (90.1%) and heart transplantation (27.3%).

Cardiac amyloid infiltration is associated with a progressive reduction in myocardial compliance and elevated left ventricular filling pressures, leading to variable degrees of HF symptoms [11,12,13]. As the disease advances, myocardial contractility becomes compromised, ultimately the reducing cardiac output and causing a gradual decline in exercise tolerance [14,15]. In our cohort of patients with ATTRv-CM linked to the His108Arg variant, we observed a high incidence of HF hospitalizations and a decline in myocardial contractility, as evidenced by a progressive decrease in LV-GLS despite a relatively stable LVEF. This trend was further corroborated by a marked increase in NT-proBNP levels over the follow-up period. In contrast, the troponin levels remained stable, suggesting that the deterioration in cardiac function is primarily driven by an impairment of myocardial relaxation and thickening, rather than a direct cardiotoxic effect of the amyloid fibrils. Additionally, the disproportionately greater rise in NT-proBNP compared to the troponin levels may be attributed to the high prevalence of atrial fibrillation in this patient population, indicating that the previous studies have shown that cardiac amyloid deposition often extends beyond the left ventricle, involving other cardiac structures such as the right ventricle and atria [16,17,18]. Notably, in our His108Arg variant population, we did not observe clinically significant right ventricular involvement, with the right ventricular function remaining normal in most cases.

Atrial arrhythmias are more prevalent in patients with CA compared to the general population [19,20]. This is most likely not only due to the progressive increase in left ventricular filling pressures due to reduced left ventricular compliance, but also the direct result of the amyloid infiltration of the atrial myocardium [21,22]. As demonstrated in our patient population, the development of AF was associated with significant symptoms and a hemodynamic compromise. In general, patients with CA have only a narrow window of an optimal heart rate range, as they, on the one hand, require a higher-than-normal heart range to compensate for the reduced stroke volume, but on the other hand, have poor tolerance to the shortened diastolic filling time caused by tachycardia. In ATTR-CM, the treatment of AF is complicated due to various factors, including the poor tolerance of betablockers, concerns of local digoxin toxicity in CA from in vitro studies, as well as high recurrence rates of AF after electrical cardioversion and ablation [23,24,25].

Previous studies have shown that even though ATTR-CM is associated with an increase in ventricular arrhythmias, ICD implantation overall does not seem to result in reduced mortality [26,27]. However, the majority of these studies did not differentiate between patients with ATTRwt and ATTRv and certainly did not account for differences in TTR mutations. The general recommendations for primary prevention ICD implantation in patients with HF, which are primarily based on LVEF, are most likely not transferable to the CA population, since these patients typically have small ventricles with numerically high LVEF, despite markedly reduced myocardial contractility, also represented by significantly reduced LV-GLS in our patient cohort. This suggests that alternative parameters of the left ventricular function would be more appropriate to aid decision making regarding ICD implantation. Furthermore, considering the heterogeneity of the ATTRv population, including variations in the degree of cardiac involvement in the disease onset and clinical course, the decision regarding ICD implantation needs to be made on an individual basis. This not only warrants evaluation by experienced amyloidosis specialists, but also requires more data which further characterize populations with different disease-causing mutations, which are currently still lacking.

Similarly, there is very little experience on the indications and timing of HTX in this population. Combined liver and heart transplantation has previously been proposed as an approach which could potentially provide a definite cure for these patients, but is associated with a high risk of postoperative complications and mortality [28,29,30,31]. Furthermore, this strategy requires a high level of surgical and postoperative expertise, which may not be available in most centers. The recent development of RNA silencers will most likely obviate the necessity for liver transplantation [32,33,34,35,36]. This could reduce the complexity of the procedure and long-term complications associated with double-organ transplantation. Furthermore, by describing the clinical trajectory of individuals with a His108Arg mutation, we may be able to better understand the onset of disease for this specific mutation and detect an ATTR-CM manifestation at an early timepoint, enabling early treatment strategies and overall improved outcomes.

Data from the global Transthyretin Amyloidosis Outcome Survey (THAOS) registry demonstrated that up to one-third of asymptomatic transthyretin gene carriers (36.2% Val30Met, 34.6% non-Val30Met) develop amyloid disease within a median of 2.2 years [37]. However, these results cannot be readily extrapolated to carriers of the His108Arg mutation as this specific variant was not represented in the THAOS cohort. In this context, it is interesting that specifically the His108Arg mutation was found in the three genotype-positive subjects with a mild tracer uptake but without other evidence of cardiac manifestation, while all phenotype-negative individuals with other mutations showed no tracer uptake on the bone scintigraphy. From this, we infer that the ideal imaging modality for the follow-up of these individuals remains unclear, as we currently do not yet fully understand the significance of the mild tracer uptake in the setting of absent pathological findings on the echocardiography and CMR. We propose that a yearly bone scintigraphy or CMR follow-up are both reasonable follow-up strategies for these patients, as both modalities are relatively safe and the costs are similar. Since CMR does not subject individuals to radiation exposure, it may be a preferable option for younger patients, while a bone scintigraphy could be used alternatively in patients with contraindications to CMR.

### Limitations

This study, focusing on the clinical trajectory of patients with the His108Arg mutation in the TTR gene, has several limitations. First, the sample size is small, reducing the generalizability of the findings and limiting the statistical power to detect significant differences in the outcomes or treatment responses. The single-center design further limits the external validity, as patient characteristics, diagnostic approaches, and treatment strategies may vary across different healthcare settings. The study’s observational design also restricts the causal inferences, as potential confounders may not be fully accounted for, despite comprehensive clinical assessments. Additionally, the study’s reliance on non-invasive diagnostics, like bone scintigraphy and echocardiography, could introduce diagnostic uncertainty, as these modalities have limitations in detecting early or subclinical disease. Genetic heterogeneity, even among patients with the same mutation, complicates the interpretation of the clinical manifestations and response to therapies, as mutation-specific progression patterns may vary. The variation in the follow-up duration and treatment regimens further adds complexity to assessing the treatment efficacy and long-term outcomes. Finally, given the rarity of the His108Arg mutation and the lack of well-established management protocols, clinical decision making remains largely individualized, underscoring the need for larger mutation-specific studies to better define optimal treatment strategies.

## 5. Conclusions

This study provides important insights into the clinical course and characteristics of patients with His108Arg-variant TTR, and contributes to the understanding of this ATTRv-CM phenotype. We highlight the heterogeneity of its manifestation and describe significant cardiac involvement, including arrhythmias and HF, which contribute to the high morbidity and reduced quality of life. As this population remains under-represented in the literature, larger studies are warranted to further clarify the natural history of the His108Arg mutation, refine risk stratification, and guide mutation-specific management.

## Figures and Tables

**Figure 1 jcm-13-07857-f001:**
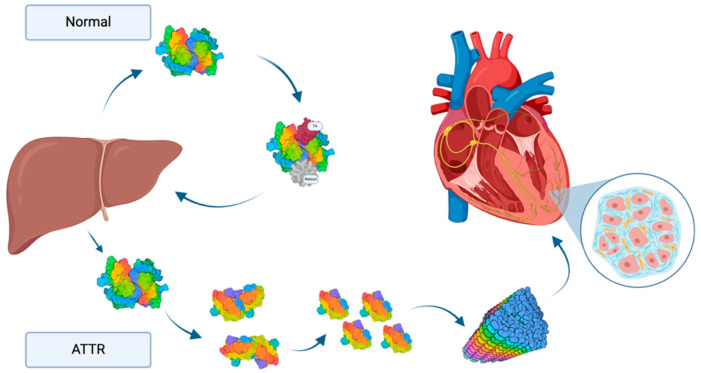
Pathophysiology of transthyretin cardiac amyloidosis shown in contrast to the physiologic function of transthyretin. Transthyretin is produced in the liver. In its stable, physiologic form, transthyretin serves as a transport protein for retinol and thyroxin (T4) which it binds in the plasma and returns to the liver. When the transthyretin protein becomes unstable, either due to age-associated (wild-type ATTR)) or genetic mutations (hereditary ATTR) it dissociates into dimers, and further into monomers. These can accumulate into amyloid fibrils, which have the tendency to deposit in various organs, including the extracellular space of the myocardium and other cardiac structures. Figure was created with biorender.com.

**Figure 2 jcm-13-07857-f002:**
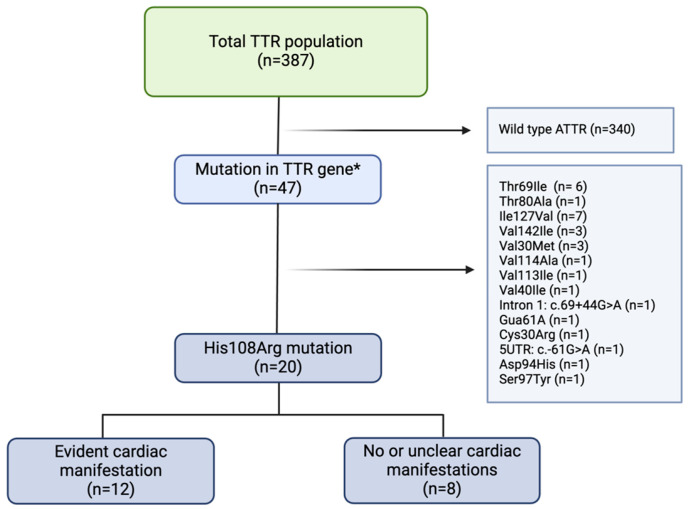
Overview of the entire transthyretin patient population and the patient selection flow. TTR indicates transthyretin. * an overview of all detected mutations can be found in the Appendix A. Figure was created with biorender.com.

**Figure 3 jcm-13-07857-f003:**
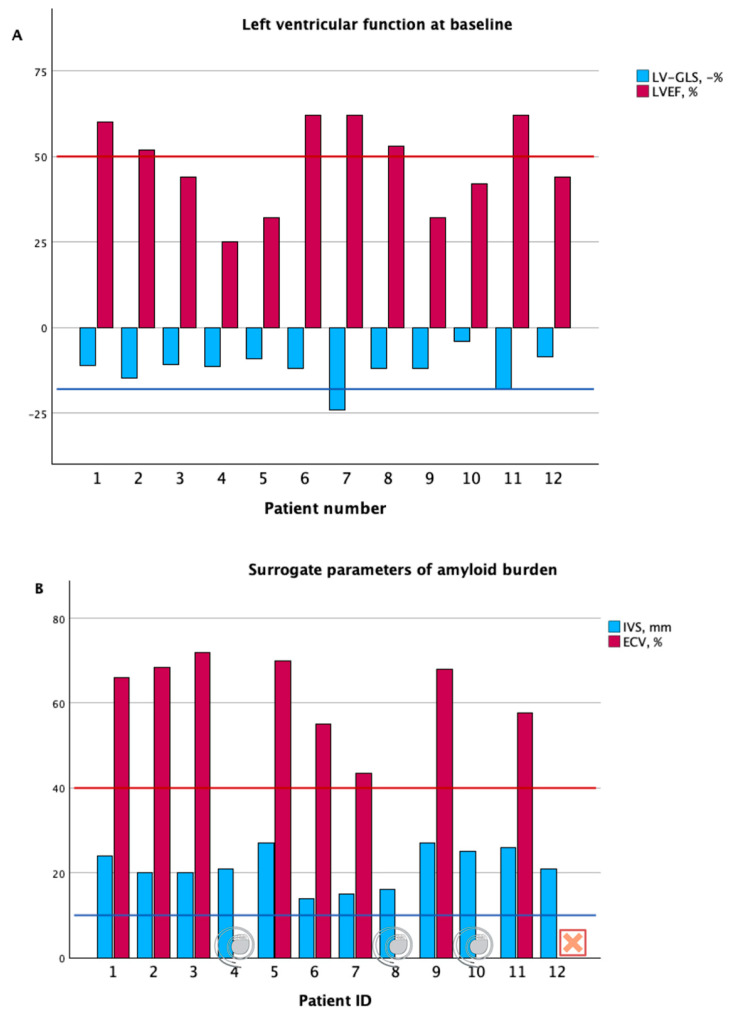
Parameters of left ventricular function (panel (**A**)) and surrogate markers of amyloid burden (panel (**B**)) in the entire transthyretin amyloid cardiomyopathy at baseline. LV-GLS, left ventricular global longitudinal strain; LVEF, left ventricular ejection fraction; ECV, extracellular volume fraction measured by cardiac magnetic resonance imaging; IVS, interventricular septum thickness measured by echocardiography. Vertical red and blue lines indicate normal cut-off values for reference. Cardiac magnetic resonance imaging for quantification of extracellular volume fraction (ECV) was not performed in Patients 4, 8, and 10 as they had a cardiac implantable device at the time of baseline evaluation. In Patient 12, cardiac magnetic resonance imaging was not performed at our center and ECV was, therefore, not available. Figure was created with biorender.com.

**Figure 4 jcm-13-07857-f004:**
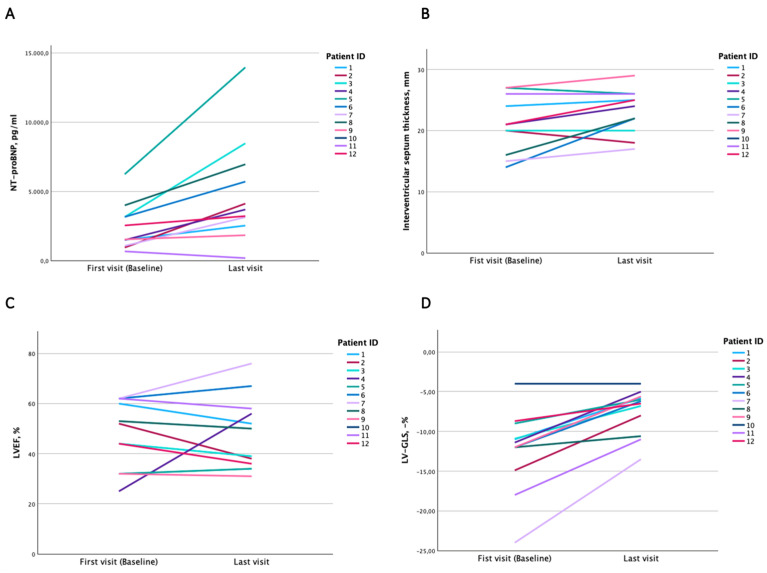
Changes in N-terminal proBNP (panel (**A**)), interventricular septum thickness (panel (**B**)), left ventricular ejection fraction (panel (**C**)) and left ventricular global longitudinal strain (panel (**D**)) over the course of the follow-up period from baseline visit to the last recorded visit. NT-proBNP indicates N-terminal pro hormone of brain natriuretic peptide; IVS is interventricular septum thickness, LVEF is left ventricular ejection fraction and LV-GLS is left ventricular global longitudinal strain.

**Figure 5 jcm-13-07857-f005:**
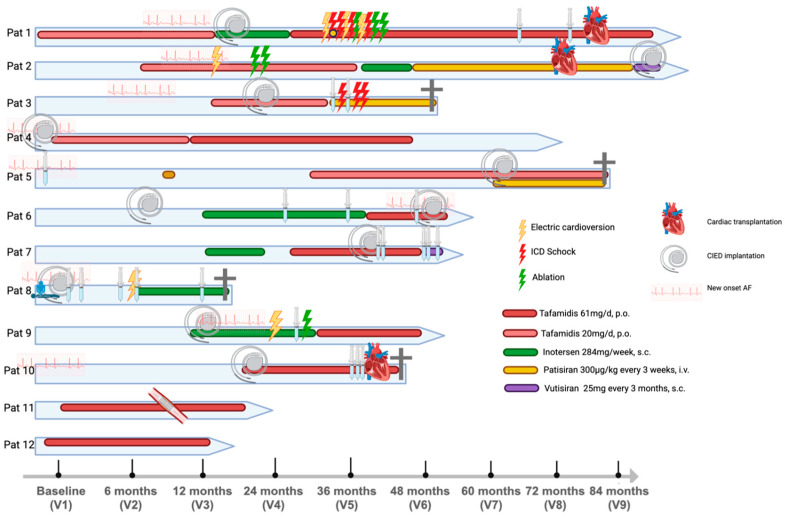
Overview of the clinical course and therapy regimens of all patients with transthyretin amyloid cardiomyopathy caused by a mutation in the His108Arg gene locus. Pat, patient and patient identifier; ICD, implantable cardiac device; V, visit number; CIED, cardiac implantable electronic device; and AF, atrial fibrillation. Figure was created with biorender.com.

**Figure 6 jcm-13-07857-f006:**
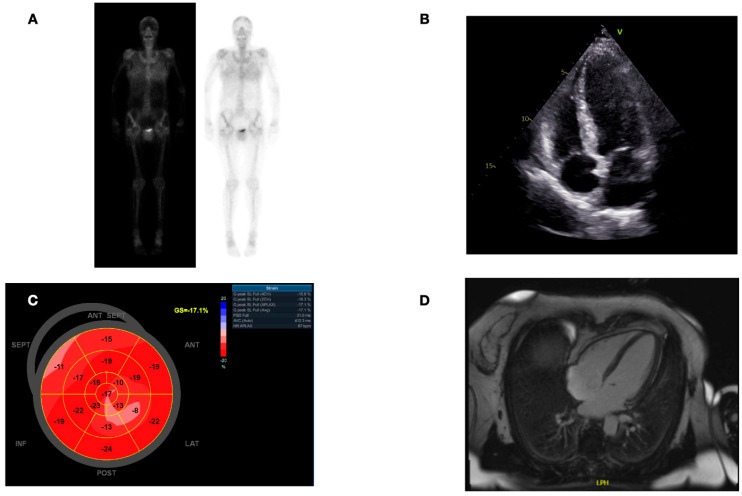
Cardiac imaging of a carrier of the His108Arg variant in the transthyretin gene without definite signs of cardiac amyloidosis. Panel (**A**) shows the bone scintigraphy images with mild tracer uptake (Perugini grade II). Panel (**B**) shows an apical four-chamber view of a transthoracic echocardiography exam without left ventricular hypertrophy (interventricular septum thickness 10 mm), normal left ventricular ejection fraction (LVEF 56%), and normal right ventricular dimensions and function. Panel (**C**) shows the bullseye display of left ventricular longitudinal strain obtained by 2D transthoracic speckle tracking echocardiography showing regions of slightly reduced strain values, without a pathognomonic “apical sparing” pattern” and a global longitudinal strain in the low normal range (−17.5%). Panel (**D**) shows of cardiac magnetic resonance imaging sequences after gadolinium administration without late gadolinium enhancement and, therefore, no evidence of cardiac amyloidosis.

**Table 1 jcm-13-07857-t001:** Characteristics of all 12 patients with transthyretin amyloid cardiomyopathy caused by a mutation in the His108Arg gene locus at time of baseline evaluation.

** *Demographic Variables and Clinical History* **	
Female	4 (33.3)
Age at time of diagnosis, years	62.3 (58.6–65.5)
Non-biopsy diagnosis	8 (66.7)
** *Comorbidities* **	
Arterial hypertension	9 (75.0)
Diabetes mellitus	1 (8.3)
Atrial fibrillation	4 (36.4)
Coronary artery disease	1 (8.3)
Polyneuropathy	5 (41.7)
** *Concomitant medication* **	
Betablockers	4 (33.3)
ACEi/ARB/ARNI	7 (58.3)
MRA	8 (66.7)
SGLT-2 inhibitor	2 (16.7)

Categorical variables are shown as numbers and percentages, continuous variables as median and interquartile range.

**Table 2 jcm-13-07857-t002:** Clinical follow-up of patients with transthyretin amyloid cardiomyopathy.

	Baseline (V1) (n = 12)	1-Year Follow-Up (V3) (n = 8)	2-Year Follow-Up (V4) (n = 7)	3-Year Follow-Up (V5) (n = 5)	4-Year Follow-Up (V6) (n = 7)	Z-Score **	*p*-Value **
Time from diagnosis to baseline visit, months	0.0 (0.0–2.4)	15.2 (14.1–17.8)	25.2 (20.6–26.8)	39.1 (38.1–39.7)	46.4 (43.2–52.0)	-	-
Body weight, kg	79.0 (70.0–122.0)	78.0 (75.0–110.0)	73.5 (66.0–118.0)	77.0 (63.0–125.0)	75.0 (64.0–101.0)	−1.40	0.161
NYHA class							
Class I	0 (0.0)	0 (0.0)	0 (0.0)	0 (0.0)	0 (0.0)	-	-
Class II	6 (54.5)	4 (50.0)	4 (57.1)	1 (20.0)	3 (42.9)	-	-
Class III	5 (45.5)	3 (37.5)	3 (42.9)	3 (60.0)	4 (57.1)	-	-
Class IV	0 (0.0)	0 (0.0)	0 (0.0)	1 (20.0)	0 (0.0)	-	-
6-MWD, m	407 (357–425)	441 (269–561)	398 (287–550)	360 (164–475)	475 (268–507)	−1.60	0.109
Systolic BP, mmHg	115 (110–120)	123 (114–135)	125 (105–130)	115 (108–125)	110 (97–120)	−1.19	0.236
Diastolic BP, mmHg	77 (70–80)	76 (74–80)	80 (70–85)	73 (65–78)	70 (65–80)	−1.07	0.286
Heart rate, bpm	75 (72–87)	90 (85–99)	83 (80–86)	65 (63–76)	69 (66–71)	−0.27	0.786
Loop diuretic dose/d, mg	30 (0–80)	20 (0–160)	60 (40–80)	60 (40–160	40 (40–160)	1.80	0.073
** *Laboratory parameters* **							
NT-pro BNP pg/mL	2033 (1278–3572)	1695 (1343–3676)	2366 (1979–3967)	3523 (3232–3606)	3183 (2136–5702)	2.76	**0.006**
Troponin T, ng/L	53 (28–82)	71 (45–79)	60 (40–80)	67 (49–68)	69 (40–102)	1.21	0.225
Serum creatinine, mg/dl	0.99 (0.72–1.12)	1.26 (0.76–1.50)	0.98 (0.79–1.21)	1.48 (1.01–1.83)	1.29 (0.88–1.59)	2.85	**0.004**
** *Echocardiography* **							
LVEDD, mm	44 (40–47)	41 (39–43)	39 (39–47)	48 (42–54)	40 (31–45)	−1.96	0.050
LVEDV, ml	79 (70–105)	75 (59–115)	81 (69–128)	75 (65–97)	75 (41–112)	−1.02	0.307
IVS, mm	21 (18–26)	22 (19–23)	21 (16–21)	23 (21–23)	22 (18–25)	2.02	**0.043**
PWT, mm	18 (17–22)	19 (19–22)	18 (16–20)	20 (16–22)	18 (17–25)	0.211	0.833
LV mass, g	410 (356–490)	373 (345–427)	361 (265–462)	523 (388–600)	418 (249–524)	−0.420	0.674
LVEF, %	48 (37–61)	58 (41–63)	44 (42–48)	52 (28–56)	40 (38–67)	−0.45	0.656
LV-GLS, -%	11.7 (13.5–10.0)	10.9 (14.0–7.0)	11.0 (11.0–8.0)	10.6 (13.7–8.7)	6.8 (8.0–5.6)	2.94	**0.003**
LA volume, mL	74 (55–124)	95 (49–115)	103 (69–119)	100 (91–101)	75 (63–113)	−5.3	0.594
Max. E-velocity, m/s	0.99 (0.90–1.2)	1.17 (1.00–1.35)	0.95 (0.79–1.2)	0.90 (0.71–1.28)	0.88 (0.70–1.09)	−0.561	0.575
Lateral e’, m/s	0.06 (0.05–0.09)	0.05 (0.05–0.05)	0.07 (0.06–0.07)	0.10 (0.10–0.10)	0.04 (0.04–0.04)	0.447	0.655
E/e’	18.4 (9.6–21.5)	21.5 (20.0–23.0)	13.4 (13.2–18.0)	7.6 (7.6–7.6))	27.3 (27.3–27.3)	−0.54	0.593
RVEDD, mm	35 (28–37)	36 (32–40)	36 (34–37)	38 (34–42)	36 (32–37)	0.05	0.959
TAPSE, mm	16 (11–21)	17 (12–20)	14 (13.19)	17 (16–19)	16 (13–19)	−2.28	0.201
RV-TDI, m/s	0.12 (0.08–0.13)	0.05 (0.08–0.12)	0.09 (0.08–0.14)	0.10 (0.09–0.11)	0.12 (0.07–0.14)	−0.635	0.526
MR-grade *	1.3 (1.0–2.0)	1.5 (1.0–2.0)	1.5 (1.5–2.0)	1.5 (1.0–2.0)	1.5 (1.0–3.0)	−0.14	0.890
TR-grade *	0.8 (0.5–1.8)	1.0 (1.0–1.5)	2.0 (1.5–2.0)	3.0 (1.5–3.0)	2.0 (1.0–3.0)	2.39	**0.017**
sPAP, mmHg	45 (33–52)	48 (37–53)	43 (30–53)	48 (37–48)	42 (35–58)	−0.24	0.799
** *Cardiac magnetic resonance* **							
LVEDV, mL	176 (157–241)	128 (81–174)	174 (166–181)	---	199 (199–199)	1.00	0.317
LVEF, %	53 (47–59)	46 (32–60)	52 (35–69)	---	34 (34–34)	−1.00	0.317
RVEDV, mL	200 (155–264)	202 (174–230)	217 (193–241)	---	241 (241–241)	1.00	0.317
RVEF, %	49 (43–51)	44 (31–56)	47 (33–61)	---	34 (34–34)	−1.00	0.317
ECV, %	68.4 (43.5–69.9)	68.4 (57.8–79.0)	62.9 (47.0–78.8)	---	78.2 (78.2–78.2)	1.34	0.180

Categorical variables are shown as numbers and percentages, and continuous variables as median and interquartile range. FU indicates follow-up; NYHA, New York Heart Association dyspnea scale; 6-MWD, six-minute walking distance; BP, blood pressure; NT-pro BNP, N-terminal pro brain natriuretic peptide; LVEDD, left ventricular end-diastolic diameter; LVEDV, left ventricular end-diastolic volume; IVS, interventricular septum thickness; PWT, posterior wall thickness; LV, left ventricle; LVEF, left ventricular ejection fraction, LV-GLS, left ventricular global longitudinal strain; LA left atrium; RVEDD, right ventricular end-diastolic diameter; RV-TDI, right ventricular tissue Doppler index; MR, mitral regurgitation; TR, tricuspid regurgitation; sPAP, systolic pulmonary artery pressure; RVEDV, right ventricular end-diastolic volume; ECV, extracellular volume. * MR and TR were graded according to a scale as a continuous variable where 0 is none, 0.5 is trace, 1.0 is mild, 1.5 is mild-to-moderate, 2.0 is moderate, 2.5 is moderate-to-severe, and 3 is severe. ** Z-score and statistical significance were calculated for first versus last visit.

**Table 3 jcm-13-07857-t003:** Overview of cardiac implantable electronic devices implanted in patients with His108Arg transthyretin amyloid cardiomyopathy.

	Type of CIED	Indication	Time from Diagnosis to Implantation	Number of ICD Shocks	Follow-Up Time
Patient 1	Single-chamber ICD	Primary prophylaxis	22 months	Appropriate: 3 Inappropriate: 1	102 months
Patient 2	Dual-chamber PM	Third-degree AV-Block after cardiac transplant	93 months	---	100 months
Patient 3	Subcutaneous ICD	Primary prophylaxis	20 months	Appropriate: 3 Inappropriate: 0	49 months
Patient 4	CRT-D	Sick-sinus syndrome and primary prophylaxis (LVEF < 35% and LBBB)	7 months prior to ATTR diagnosis	Appropriate: 0 Inappropriate: 0	76 months
Patient 5	CRT-D	Bradycardic AF and primary prophylaxis (LVEF < 35%, LBBB)	65 months	Appropriate: 0 Inappropriate: 0	82 months
Patient 6	Single-chamber ICD --- CRT-D upgrade	Primary prophylaxis --- High-grade AV block and pacing dependency with progressive LV-dysfunction and symptoms	8 months --- 54 months	Appropriate: 0 Inappropriate: 0	62 months
Patient 7	Dual-chamber ICD	Primary prophylaxis	41 months	Appropriate: 0 Inappropriate: 0	60 months
Patient 8	Dual-chamber ICD	Secondary prophylaxis after survived SCD	2 months prior to ATTR diagnosis	Appropriate: 0 Inappropriate: 0	20 months
Patient 9	Dual-chamber ICD	Primary prophylaxis	12 months	Appropriate: 0 Inappropriate: 0	57 months
Patient 10	Single-chamber ICD	Primary prophylaxis	21 months	Appropriate: 0 Inappropriate: 0	44 months
Patient 11	None	---	---	---	23 months
Patient 12	None	---	---	---	18 months

CIED, cardiac implantable electronic device; ICD, implantable cardiac defibrillator; PM, pacemaker; CRT-, cardiac resynchronization therapy with defibrillator function; LVEF, left ventricular ejection fraction; LBBB, left bundle-branch block; AV-Block, atrioventricular block; SCD, sudden cardiac death; ATTR, transthyretin amyloidosis.

**Table 4 jcm-13-07857-t004:** Procedures and outcomes of the entire population with a His108Arg mutation causing transthyretin amyloid cardiomyopathy over a median follow-up of 5.7 (4.7–8.4) years.

	Death	Cardiac Transplantation	Electric Cardioversions	Ablation Therapies	Worsening of Heart Failure
**Patient 1**	No	Yes	3	3	2
**Patient 2**	No	Yes	1	0	2
**Patient 3**	Yes, non-cardiac	No	0	0	2
**Patient 4**	No	No	0	0	0
**Patient 5**	Yes	No	0	0	1
**Patient 6**	No	No	0	0	3
**Patient 7**	No	No	0	0	6
**Patient 8**	Yes, cardiac	No	1	0	5
**Patient 9**	No	No	1	1	1
**Patient 10**	Yes, cardiac	Yes	0	0	4
**Patient 11**	No	No	0	0	0
**Patient 12**	No	No	0	0	0

## Data Availability

Dataset available on request from the authors.

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
