# Peer review of "His108Arg Transthyretin Amyloidosis—Shedding Light on a Distinctively Malignant Variant"

_jcm, 2024, doi:10.3390/jcm13247857_

Round 1
Reviewer 1 Report
Comments and Suggestions for Authors
The presented article is undoubtedly of great interest, and the ability to provide such a detailed and prolonged follow-up for a rarely described mutation is remarkable.
However, some minor improvements can be made regarding:
- The numbering of some figures is incorrect: Figure 4 (pages 8-9) appears before Figure 3 (page 10).
- In Figure 5, what is the time scale used? Please specify!
- It would be appropriate to provide a more detailed description of the neurological phenotype of patients with polyneuropathy. What type of polyneuropathy do they present? Did it appear before or after the cardiomyopathy? What is the severity of the neuropathy?
- It should be specified whether any of the 20 described patients are related to others within the population.
Author Response
Thank you for giving us the opportunity to submit a revised version of our manuscript. We have tried to carefully answer all comments and remarks raised by the reviewers and believe that through this process the manuscript has substantially improved.
Reviewer comment 1: The numbering of some figures is incorrect: Figure 4 (pages 8-9) appears before Figure 3 (page 10). In Figure 5, what is the time scale used? Please specify!
Response to comment 1: Thank you very much for pointing this out. We apologize for the inconsistencies and have updated the numbering of the figures so that now “Figure 3” is renamed to “Figure 4” (page 10) and “Figure 4” is now “Figure 3” (page 8-9).
Furthermore, we have updated figure 5 to add the time scale. The updated figure now shows as follows:
Figure 5- Overview of the clinical course and therapy regimens of all patients with transthyretin amyloid cardiomyopathy caused by a mutation in the His108Arg gene locus.

Pat indicates patient and patient identifier; ICD, implantable cardiac device; V, visit number; CIED, cardiac implantable electronic device and AF, atrial fibrillation.
Figure was created with biorender.com
Reviewer comment 2: It would be appropriate to provide a more detailed description of the neurological phenotype of patients with polyneuropathy. What type of polyneuropathy do they present? Did it appear before or after the cardiomyopathy? What is the severity of the neuropathy?
Response to comment 2: Thank you for addressing the issue of neurological involvement. Even though the present study focusses on ATTR cardiomyopathy, we very much agree that cardiac involvement can- and should not be seen in isolation and that describing the presence and degree of polyneuropathy in this cohort would substantially improve the manuscript overall. We therefore added a table to the supplementary materials (table S2) that shows the estimated onset of polyneuropathy symptoms, as well as the results of the neurological evaluation, as available in hospital records. Furthermore, we adapted the results section, subsection “3.2 Patient characteristics at the time of clinical diagnosis” and now describe neurological involvement in more detail and reference the newly added table in the text.
Changes made to subsection 3.2 of the results:
Polyneuropathy (PNP) was confirmed in 5 patients at the time of baseline evaluation. In only one patient, PNP had been diagnosed prior to the diagnosis of ATTR-CM (patient 5). This patient was significantly impaired by PNP symptoms and the neurological diagnosis prompted cardiac work-up. The remaining patients with evidence of PNP on nerve conduction testing experienced only mild symptoms. A detailed description of neurological manifestations is shown in Table S2 in the appendix.
Table S2- Overview of neurologic involvement in the patient population with positive cardiac phenotype.
|
|
Initial, predominant manifestation |
Previous CTS-surgery |
Nerve conduction test |
Clinical presentation |
|
Patient 1 |
Cardiac |
Unilateral |
Bilataral senosmotory PNP of UE and LE |
Dysesthesia in both UE |
|
Patient 2 |
Cardiac |
No |
No PNP |
-- |
|
Patient 3 |
Cardiac |
No |
No PNP |
-- |
|
Patient 4 |
Cardiac |
No |
Bilataral senosmotory PNP of LE |
No significant symptoms |
|
Patient 5 |
Neurological |
No |
Bilataral senosmotory PNP of UE and LE |
Impaired fine motor skills and gait instability |
|
Patient 6 |
Cardiac |
Bilateral |
Bilateral senosmotory PNP of UE and LE |
Dysaesthesia of both UE |
|
Patient 7 |
Cardiac |
Bilateral |
Bilateral sensomotory PNP of UE < LE |
Restless leg symptoms, dysaesthesia of both UE and LE |
|
Patient 8 |
Cardiac |
No |
Bilateral sensomotory PNP of UE and LE |
Dysesthesia in both UE |
|
Patient 9 |
Cardiac |
Bilateral |
Bilateral sensomotory PNP of UE and LE |
Dysesthesia in both UE |
|
Patient 10 |
Cardiac |
No |
Bilateral, predominantly sensory PNP of LE |
No significant symptoms |
|
Patient 11 |
Cardiac |
Unilateral |
Bilateral sensomotory PNP of UE and LE |
Dysaethesia in left UE |
|
Patient 12 |
Cardiac |
Bilateral |
LE-dominant, bilateral sensomotory PNP |
No significant symptoms |
.
Reviewer comment 3: It should be specified whether any of the 20 described patients are related to others within the population.
Response to comment 3: We thank the reviewer for this comment. The respective adaptations have been made. In fact, the 20 patients in the phenotype-positive cohort are included in table S1, however we now realize, that this is not evident, as it is not clearly stated and we have therefore updated the table for more clarity. Furthermore, we have added a statement to the results, subsection “3.1 Patient population and mode of diagnosis”, which describes the family relationship within the study population. Unfortunately, our clinical records do not contain information on how exactly these individuals are related.
The following statement was added to the results section, “3.1 Patient population and mode of diagnosis”:
In patients with His108Arg-associated ATTR-CM, the median age at diagnosis was 62.3 years (IQR 58.6 – 65.5). The ATTR-CM population consisted of 8 males and 4 females, while the 8 individuals with verified His108Arg mutation without definite cardiac manifestations were 5 females and 3 males. Notably, we identified two family clusters; patients 2, 11, and 12 were related, as well as patients 6, 7 and 8 (table S1). Patient characteristics at the time of baseline evaluation are shown in table 1.
Table S1 after revision:
Table S1- Overview of detected transthyretin mutations during patient and family screening
|
Sex |
Age* |
Mutation |
Phenotype |
DPD scan |
|
|
Female (Patient 1) |
56 |
His108Arg |
Positive |
Perugini III |
|
|
Male |
40 |
His108Arg |
Negative |
Perugini 0 |
|
|
Male (Patient 2) |
56 |
His108Arg |
Positive |
Perugini III |
|
|
Female |
59 |
His108Arg |
Negative |
Perugini II |
|
|
Male (Patient 11) |
66 |
His108Arg |
Positive |
Perugini III |
|
|
Male |
35 |
His108Arg |
Negative |
Perugini 0 |
|
|
Male (Patient 12) |
60 |
His108Arg |
Positive |
Perugini II |
|
|
Female (Patient 3) |
61 |
His108Arg |
Positive |
Perugini III |
|
|
Male (Patient 4) |
70 |
His108Arg |
Positive |
Perugini III |
|
|
Male |
33 |
His108Arg |
Negative |
Perugini 0 |
|
|
Female |
40 |
His108Arg |
Negative |
Perugini 0 |
|
|
Male (Patient 5) |
62 |
His108Arg |
Positive |
Perugini III |
|
|
Female (Patient 7) |
60 |
His108Arg |
Positive |
Perugini III |
|
|
Male (Patient 6) |
62 |
His108Arg |
Positive |
Perugini III |
|
|
|
Male (Patient 8) |
64 |
His108Arg |
Positive |
Perugini III |
|
|
Female |
70 |
His108Arg |
Negative |
Perugini I |
|
|
Female |
59 |
His108Arg |
Negative |
Perugini 0 |
|
Male (Patient 10) |
48 |
His108Arg |
Positive |
Perugini III |
|
|
Female (Patient 9) |
68 |
His108Arg |
Positive |
Perugini III |
|
|
Female |
69 |
His108Arg |
Negative |
Perugini 0 |
|
|
Male |
68 |
Thr69Ile |
Positive |
Perugini III |
|
|
Male |
67 |
Thr69Ile |
Positive |
Perugini III |
|
|
Female |
35 |
Thr69Ile |
Negative |
Perugini 0 |
|
|
Male |
30 |
Thr69Ile |
Negative |
Perugini 0 |
|
|
Female |
42 |
Thr69Ile |
Negative |
Perugini 0 |
|
|
Male |
60 |
Thr69Ile |
Positive |
Perugini III |
|
|
Female |
72 |
Ile27Val |
Positive |
Perugini III |
|
|
Male |
60 |
Ile127Val |
Positive |
Perugini III |
|
|
Female |
77 |
Ile127Val |
Positive |
Perugini III |
|
|
Female |
42 |
Ile127Val |
Negative |
Perugini 0 |
|
|
Female |
33 |
Ile127Val |
Negative |
Perugini 0 |
|
|
Male |
76 |
Ile127Phe |
Positive |
Perugini III |
|
|
Male |
59 |
Ile127Phe |
Negative |
Perugini 0 |
|
|
Male |
80 |
Val142Ile |
Positive |
Perugini III |
|
|
Male |
45 |
Val142Ile |
Negative |
Perugini 0 |
|
|
Male |
45 |
Val142Ile |
Negative |
Perugini 0 |
|
|
Male |
86 |
Val30Met |
Positive |
Perugini III |
|
|
Male |
81 |
Val30Met |
Positive |
Perugini III |
|
|
|
Male |
77 |
Val30Met |
Positive |
Perugini III |
|
Female |
68 |
Val114Ala |
Positive |
Perugini III |
|
|
Female |
75 |
Val113Leu |
Positive |
Perugini III |
|
|
Male |
77 |
Val40Ile |
Positive |
Perugini III |
|
|
Male |
78 |
Intron 1: c.69+44G>A)* |
Positive |
Perugini III |
|
|
Male |
68 |
Thr80Ala |
Positive |
Perugini III |
|
|
Female |
87 |
Cys30Arg |
Positive |
Perugini III |
|
|
Male |
80 |
5UTR: c.-61G>A * |
Positive |
Perugini III |
|
|
Male |
59 |
Asp94His |
Negative |
Perugini 0 |
|
*In patients with positive cardiac phenotype, age at time of diagnosis is given, while in phenotype negative patients, age at time of screening is shown.
** Mutation in the transthyretin gene of unknown significance. Rows with thick borders indicate that individuals are blood relatives.
Reviewer 2 Report
Comments and Suggestions for Authors
In this work, Binder et al. collected clinical data and described the characteristics and clinical course of patients carrying the transthyretin gene (TTR) His108Arg variant in the context of transthyretin amyloid cardiomyopathy (ATTRv-CM). The study covered baseline evaluation, involvement of heart failure and arrhythmia, treatment, and outcomes, which provides a relatively comprehensive characterization of the current understanding of TTR His108Arg variant-associated ATTRv-CM. While the population of this study is limited, it provides a valuable contribution to investigating this understudied TTR variant. However, there are still a few minor points that need to be addressed before the acceptance for publication:
1. In line 269, the NT-proBNP significantly increased compared to baseline, while troponin only slightly increased. Could you explain more about this discrepancy?
2. In line 299, could you explain more here to define what appropriate and inappropriate ICD shocks are?
3. In line 340, could you provide the patient number for this case? Is the tolerance of Inotersen associated with any specific cardiac parameters or disease progression? It would be helpful to check why this patient can tolerate Inotersen.
Author Response
Thank you for allowing us to submit a revised version of our manuscript. We have tried to carefully answer all comments and remarks raised by the reviewers and believe that through this process the manuscript has substantially improved.
Reviewer comment 1: In line 269, the NT-proBNP significantly increased compared to baseline, while troponin only slightly increased. Could you explain more about this discrepancy?
Response to comment 1: We appreciate the reviewer’s comment regarding the discrepancy between the significant increase in NT-proBNP levels and the relatively minor increase in troponin levels.
In our cohort, several patients experienced worsening heart failure during follow-up, likely contributing to the significant rise in NT-proBNP levels. Conversely, troponin T levels, which primarily indicate myocardial injury, may have remained relatively stable as TTR amyloid is not directly cardiotoxic (as opposed to light-chain amyloid). We observerd that patients in this cohort commonly developed atrial fibrillation, which typically leads to a substantial increase in NT-proBNP, whiel Troponin levels are less affected. The modest increase in troponin T levels could reflect progressive structural changes and increased wall stress but not acute or chronic myocyte necrosis.
We also acknowledge that the small patient number in our study may have limited our ability to detect subtle trends in troponin levels, as in general NT-pro BNP levels are assessed more frequently than troponin levels. Additionally, variations in laboratory work-up, such as timing of blood draw in relation to volume status and acute clinical events, could contribute to minor inconsistencies in biomarker trends, especially in such a small cohort. Finally, we note that NT-proBNP trends in TTR amyloidosis are well-documented as a more robust indicator of disease progression compared to troponin. This finding has been corroborated in prior studies, which highlight the use of NT-proBNP as a prognostic marker for worsening heart failure and mortality in this patient population (Oghina S et al. J Clin Med. 2021, https://doi.org/10.3390/jcm10214868)
We have added the following short paragraph to the discussion section of the revised manuscript:
This trend was further corroborated by a marked increase in NT-proBNP levels over the follow-up period. In contrast, troponin levels remained stable, suggesting that the deterioration in cardiac function is primarily driven by an impairment of myocardial relaxation and thickening, rather than a direct cardiotoxic effect of amyloid fibrills. Additionally, the disproportionately greater rise in NT-proBNP compared to troponin levels may be attributed to the high prevalence of atrial fibrillation in this patient population.
Reviewer comment 2: In line 299, could you explain more here to define what appropriate and inappropriate ICD shocks are?
Response to comment 2: Certainly, we agree that this definition should be explained in the manuscript to avoid misinterpretations. We have therefore added a sentence that elucidates the definition of appropriateness of ICD shocks. Subsection “2.5, Definition of outcomes” has been changed in the manuscript and now reads as follows:
The primary clinical outcomes were defined as cardiac death and worsening of HF. Cardiac death was classified as any death directly attributable to cardiac causes, including arrhythmic events, progressive HF, or sudden cardiac arrest, in line with established criteria. Worsening of HF was characterized by the need for hospitalization due to HF symptoms, an escalation in diuretic treatment, or a documented decline in functional status requiring medical intervention. In patients who had an intracardiac implantable defibrillator device (ICD) at baseline or received such a device during follow-up, we also assessed the number and appropriateness of shocks from the respective device during the observation period. Appropriate ICD-shocks were defined as a therapeutic intervention delivered by the device in response to a detected life-threatening ventricular arrhythmias, such as ventricular tachycardia or ventricular fibrillation.
Reviewer comment 3: In line 340, could you provide the patient number for this case? Is the tolerance of Inotersen associated with any specific cardiac parameters or disease progression? It would be helpful to check why this patient can tolerate Inotersen.
Response to comment 3: Thank you for appreciating this. We also found it clinically interesting, that tolerance was poor in some patients. The patient who initially tolerated Inotersen well and then had a reaction approximately 7 months later was patient 9. After reviewing the baseline and follow-up table of patient characteristics, as well as the raw data, we noticed that this patient had very pronounced LV hypertrophy as well as LVEF on the lower spectrum compared to the remaining population, indicating more severe disease. We believe that the reason why patient 9 initially tolerated the injections and then had a reaction cannot be explained by the parameters available to us and would more likely be associated with immunological factors or potentially also a difference in the self-administration of pre-medication. Since this drug was self-administered by the patient at the time, detailed information on the circumstances is lacking and no in-depth evaluation of potential causes of the drug reaction was performed, as it was considered safest by our team simply to switch to an alternative therapy regime in this specific patient. We have specified the patient number in the revised version of the results, subsection “3.5. Medical management and TTR-specific treatment regimens”:
Of the 6 patients who were prescribed Inotersen, 4 patients discontinued the injections due to adverse effects. Interestingly, all patients initially tolerated injections well, but developed side effects including chills or fevers as well as gastrointestinal side effects over the course of several injections which led to a discontinuation of Inotersen. Patient 9 experienced a severe reaction which required hospitalization immediately following an injection after 7 months of therapy. Patient 6 developed thrombopenia and recurrent nose bleeds and bruising, which resolved after Inotersen was suspended. Of note, this patient had significantly impaired renal function and a creatinine clearance of 22ml/min/m2. The only patient who tolerated Inotersen well was eventually switched to Patisiran due to disease progression (patient 2).